# Distribution of Prophages in the *Oenococcus oeni* Species

**DOI:** 10.3390/microorganisms9040856

**Published:** 2021-04-16

**Authors:** Olivier Claisse, Amel Chaïb, Fety Jaomanjaka, Cécile Philippe, Yasma Barchi, Patrick M. Lucas, Claire Le Marrec

**Affiliations:** Unité de Recherche Œnologie, Bordeaux INP, University of Bordeaux, INRAE, ISVV, F-33882 Bordeaux, France; olivier.claisse@u-bordeaux.fr (O.C.); bounoise68@hotmail.fr (A.C.); jaomazava@yahoo.fr (F.J.); cecilemphilippe@gmail.com (C.P.); barchiyasma34@gmail.com (Y.B.); patrick.lucas@u-bordeaux.fr (P.M.L.)

**Keywords:** bacteriophage, *Oenococcus oeni*, malolactic fermentation, integrase, recombination

## Abstract

*Oenococcus oeni* is the most exploited lactic acid bacterium in the wine industry and drives the malolactic fermentation of wines. Although prophage-like sequences have been identified in the species, many are not characterized, and a global view of their integration and distribution amongst strains is currently lacking. In this work, we analyzed the complete genomes of 231 strains for the occurrence of prophages, and analyzed their size and positions of insertion. Our data show the limited variation in the number of prophages in *O. oeni* genomes, and that six sites of insertion within the bacterial genome are being used for site-specific recombination. Prophage diversity patterns varied significantly for different host lineages, and environmental niches. Overall, the findings highlight the pervasive presence of prophages in the *O. oeni* species, their role as a major source of within-species bacterial diversity and drivers of horizontal gene transfer. Our data also have implications for enhanced understanding of the prophage recombination events which occurred during evolution of *O. oeni*, as well as the potential of prophages in influencing the fitness of these bacteria in their distinct niches.

## 1. Introduction

The wine-making process starts with the selection of the fruit and the fermentation of sugars into alcohol by yeasts. In most red and dry white wines, malolactic fermentation (MLF), which reduces acidity, increases microbial stability and creates good-quality grape wine, is also a required step. MLF is largely driven by the lactic acid bacterium (LAB) *Oenococcus oeni* [1]. This bacterium is rarely detected in the natural environment, including the surface of grape berries [2], and its population only increases after crushing. At this step, *O. oeni* is part of a complex microbiota that still comprises different LAB. Wine conditions become progressively harsher for most bacteria, except for *O. oeni*, which becomes the sole detectable LAB species isolated from wine when MLF typically occurs, and is indeed the best adapted species to the combined inhibitory effects of low pH, low oxygen, high alcohol, polyphenolic compounds and sulfur dioxide contents [2]. However, intra-species diversity is reported in the *O. oeni* species and large phenotypic differences among indigenous strains corresponding to variable capacities to withstand stringent wine compositions and impact aromas have been repeatedly observed. This has become the focus of more detailed studies, with the aim to explain microbial phenotypes on a genomic scale. To this aim, collections of strains have been assembled worldwide, from all wine regions, and also from other niches supporting the growth of *O. oeni* such as apple cider and kombucha, a fermented tea containing very little alcohol [2]. As from 2005 [3], an enormous amount of information has been progressively obtained from whole genome sequencing, showing that *O. oeni* is a highly diverse species, with genetic adaptation to fermented beverages [4,5,6,7]. Such a specialization to life in wine is visible in the small (1.6 to 2.2 Mbp) genome of *O. oeni* because the species has lost many metabolic abilities due to its adaptation to an environment rich in amino acids, vitamins and other nutrients [7]. The population structure revealed by MLST and phylogenomic analyses currently recognizes four main phylogenetic groups (referred to as groups A, B, C and D) [7,8,9]. Group A exclusively contains wine strains, and is divided into subgroups which are associated with certain types of wine, such as white wines of Burgundy or Champagne, supporting the idea that group A strains are the most domesticated to wine [10]. Groups B and C contain both cider and wine strains [7]. More recently, strains originating from kombucha tea were shown to form a fourth group named phylogroup D, which may have preceded the advent of other groups [9].

A key factor in the rapid evolution and adaptation of the strains is undoubtedly the hypermutability of the *O. oeni* species [11]. As reported for a few other species, the absence of the ubiquitous mismatch repair (MMR) genes *mutS* and *mutL* results in the accumulation of spontaneous errors in DNA replication, or in reduced stringency in recombination, thus generating high levels of polymorphism, increasing adaptation to environmental fluctuation [12]. The mutator phenotype also promotes horizontal distribution of essential genes for survival, and therefore increases the genetic innovation rate, eliciting environmental adaptation [12]. The species is also devoid of a CRISPR-Cas system, the bacterial immunity defense mechanism against foreign DNA [2,3,4,5,6,7,8,9,10,11]. Given these peculiar characteristics, it is therefore not surprising that mobile genetic elements such as insertion sequences (ISs) [13], plasmids [14] and DNA of (pro)phage origin [15] have been shown to constitute a significant portion of the *O. oeni* pangenome. The observed prevalence of prophages in bacterial genomes is indeed in agreement with previous cultivation-based assessments of lysogeny reported in the species [16].

Additional insights into the diversity of O. oeni prophages was obtained through the analysis of two major constituents of the site-specific recombination (SSR) unit, corresponding to the phage integrases and attachment sites [4,17]. Alignment of the integrase protein sequences was shown to cluster prophages into four major groups (Int_A_ to Int_D_), which were related to the integration site in the host chromosome [17]. Thus far, all groups, except Int_C_, contain mitomycin C-inducible prophages. Accordingly, no Int_C_ phage has been isolated as free phage particles during field campaigns targeting samples from grapes, wines and materials [17,18].

The abundant literature on diverse ecosystems fosters further effort to generate deeper knowledge of lysogeny in the *O. oeni* species. In particular, there is a convergence of case studies towards a role of prophages in providing their host bacterium with ecologically significant traits [19,20]. The later include immunity against specific phage attack [21], virulence [22] or specific adhesion properties through the modification of the cell surface matrix [20,23], thereby improving bacterial survival under changing conditions. Ecosystem-level consequences of lysogeny have also been documented. For example, prophages modulate the interactions between microbial symbionts and their insect or human hosts [24,25]. The current trend toward understanding phage–host interactions on a global scale also makes a valuable contribution on how phages influence bacterial diversification and ecotype formation [26,27]. The deciphering of the mechanisms of ecological divergence in the *Oenococcus* genus is highly topical, as three novel species have recently been proposed during exploration of different fermented beverages [2]. This has expanded our knowledge of the horizons of their habitat to other niches such as alcohol production from sugar cane for *O. alcoholitolerans* [28], shochu distillates for *O. kitaharae* [29], ciders for *O. oeni* and *O. sicerae* [30] and kefirs for *O. sicerae* [31]. Thanks to the recent increase in the number and diversity of sequenced genomes, we investigated the distribution, organization and insertion of prophages in *O. oeni* and its related species. Our data will help in understanding the evolutionary trajectories of strains and their phages in fermented beverages, and can be further used to explore whether prophages play a role during the adaptation of their host to wine-making conditions.

## 2. Material and Methods

### 2.1. Data Collection and Identification of Candidate Prophage Sequences in Complete Genomes of O. oeni

A total of 231 publicly available genome sequences were obtained from the National Center for Biotechnology Information NCBI, https://www.ncbi.nlm.nih.gov (accessed on 2 January 2021) (Appendix A). Corresponding strains have been collected worldwide from different fermented beverages (red, dry and sweet white wines, sparkling wines and more recently cider and kombucha tea). Candidate prophage-like elements were identified using the RAST pipeline which provides an automated approach to phage genome annotation [32]. As suggested by others [33], we performed a manual inspection of the sequences for the presence of signature sequences: attachment sites (*att*), gene(s) encoding integrase(s), terminases(s), transposases(s), genes coding for structural viral proteins and the sequences of prophage integration sites. In particular, a multi-FASTA file containing the four publicly available integrase protein sequences described among temperate oenophages was used as a reference to blast the draft genomes. In addition, the prophage search tool PHASTER was also used [34].

### 2.2. Comparisons and Phylogenetic Analyses

For specific ORF phylogenetic analyses, protein sequences were aligned using ClustalOmega at https://toolkit.tuebingen.mpg.de/ (accessed on 2 January 2021).

Phage genome comparisons were conducted using the Genome-BLAST Distance Phylogeny (GBDP) method using VICTOR under settings recommended for prokaryotic viruses [35] at: http://ggdc.dsmz.de/phylogeny-service.php (accessed on 2 January 2021). The resulting intergenomic distances were used to infer a balanced minimum evolution tree with branch support via FASTME including SPR postprocessing for each of the formulas D0, D4 and D6, respectively. Branch support was inferred from 100 pseudo-bootstrap replicates each. Trees were rooted at the midpoint and visualized with FigTree [36] as already described [37].

### 2.3. PCR

Bacterial strains CRBO11105 and CRBO14210 (accession numbers LKSR01 and LKRV01) were obtained from the Centre de Ressources Biologiques Oenologiques (CRB Oeno; ISVV, Villenave d’Ornon, France). Genomic DNAs from were extracted from cultures by using the Whatman^®^ FTA Clone Saver card technology (Whatman, Sigma-Aldrich, France). PCR amplification reactions were achieved in a 25 µL final volume with 0.2 µM of each primer. The Taq 5X Master Mix kit (New England, Biolabs, Evry, France) was used according to the manufacturer’s recommendations. A BioRad i-Cycler was used for amplification. Bacterial DNA was introduced in the reaction mixture as 1.2 mm FTA discs which were punched out of the center of FTA cards using a Uni-Core Punch (Qiagen Cat. No. WB100028, Whatman^®^, Fisher Scientific, Schwerte, Germany) and transferred to PCR tubes. The pair of primers targeting the *attB_B_* site was described earlier [17].

### 2.4. Induction with MC

For the induction of phages, mitomycin C was used as the inducing agent (1 μg/mL), with bacterial culture on modified MRS. Overnight cultures were diluted 10-fold in 10 mL of fresh broth, grown to an optical density at 600 nm (OD_600_) of 0.2 to 0.3 prior to the addition of inducing agent and incubated for 24 h. OD_600_ was measured periodically.

## 3. Results and Discussion

### 3.1. The Genome of O. oeni Is Replete with Putative Prophages

The systematic interrogation of 231 complete genomes assessed from GenBank (NCBI) resulted in the discovery of seemingly intact prophages in 134 strains of *O. oeni* (58% of the bacterial genomes studied) (Table 1).

Lysogens originated from all types of beverages (wines, ciders, kombucha), and belonged to all four lineages described in *O. oeni*, including the less numerously represented phylogroups associated with cider (B and C) and kombucha (D). Lysogens mostly harbored a single prophage (64.2%; *n* = 86). The 48 poly-lysogens contained two (*n* = 43) or three distinct prophages (*n* = 5) (Table 1).

This first survey yielded a total of 187 prophages. They showed a genome length ranging from 35 kb to 46.2 kb which, on average, and comprised between 2 and 6.7% of the host chromosome. All prophages encoded identifiable phage-specific functions such as integrases and terminases and tail-associated, portal-associated and lysis-associated proteins. The presence of a large number of predicted proteins characteristic of tailed phages (e.g., terminase, tape measure protein, tail formation and baseplate-related proteins) is consistent with previous observations by transmission electronic microscopy showing that oenophages display the Siphoviridae morphology [16,17,18]. Prophages also demonstrated well-conserved patterns in genome organization. Starting with the gene encoding the integrase, the following order was observed: lysogeny module followed by modules for replication, DNA packaging, head morphogenesis, tail, lysis and finally lysogenic conversion.

During a previous screening and analysis of prophages in a set of 42 publicly available genomes, we reported the existence of four distinct groups of temperate oenophages (Int_A-D_) [17]. The properties shared by all known members of a group included: (1) high identity of the integrase sequences (>98% at the amino acid level) and (2) tropism for a specific bacterial attachment site corresponding to the 3′ end of a tRNA gene [17]. To get a broader view, we extended our in silico analyses to the 187 newly retrieved prophages. The clustering of the majority of their integrase sequences into the four proposed groups (96%, *n* = 180) was confirmed. All identified Int_A_, Int_C_ and Int_D_ integrases were observed to preferentially drive prophage integration into their expected cognate site, encompassing the 3′ end of the *tRNA^Glu^^-^^0506^* (*attB*_A_), *tRNA^Lys^^-^^0685^* (*attB*_C_) and *tRNA^Leu^^-^^1359^* (*attB*_D_) sequences (gene numbers are those present in the PSU1 reference strain), respectively (Figure 1).

Intriguing features were observed for some Int_B_ prophages. First, we confirmed that integration at the previously described *tRNA^Leu^^-^^0851^* (*attB*_B_) site is widespread in the group, as this applied to 78% of the Int_B_ prophages [17]. In contrast, 22% showed an unusual localization outside the *attB_B_* site. Instead, recombination was shown to occur at the 3′ end of the *tRNA^Leu^^-^^0923^* gene, located 66 kb away (Figure 1). As this situation occurred in strains CRBO11105 and CRBO14210 from our strain collection, experimental verification of such an unusual integration site was done by PCR. We observed the failure to generate a PCR product using the bacterial chromosomal DNAs as a template and PCR primers flanking the *tRNA^Leu^^-^^0923^* gene, while an amplicon corresponding to the *attL* junction between bacterial and phage DNA was amplified in both strains. The Int_B_ phages were therefore integrated at the *tRNA^Leu^^-^^0923^* site in strains CRBO11105 and CRBO14210. Accordingly, the intact *attB_B_* region spanning the *tRNA^Leu^^-^^0851^* gene was amplified, confirming the absence of prophage at this site in both strains. It is therefore shown that a few Int_B_ prophages become integrated at a 63 bp secondary *att* site containing the 3′ end of the *tRNA^Leu^^-^^0923^* gene in *O. oeni*. This finding is in accordance with previous data suggesting that the Int_B_ phage 10MC could use the *tRNA^Leu^^-^^0923^* gene as a secondary site during lysogenization assays of the MCW strain [38].

Our genomic survey also indicated that seven of the 187 newly retrieved prophages harbored integrases with lower pairwise sequence identity (60–92%) with the existing Int_A-D_ proteins (Table 2).

Three prophages harbored a 100% conserved integrase sequence, which shared 60 to 69% aa identity with the sequences affiliated to the four existing groups. This novel integrase, named Int_E_, was associated with a 39.5 kb and a 38.9 kb prophage in the cider strains CRBO1384 and CRBO1389, respectively. It was also harbored by a slightly larger 46.3 kb prophage in strain AWRIB663 originating from wine. All three Int_E_ prophages targeted a 11 bp *attB*_E_ core sequence to lysogenize their host, which consisted of the 3′ end of the unique *cysteinyl tRNA*^0816^ gene in *O. oeni* [3]. The use of a *tRNA^Cys^* as a target for phage integration is rare amongst phages of LAB, but has been reported for a few phages such as TPW22 [39] and PH15 [40].

A second novel integrase, named Int_F_, was associated with a 41.7 kb resident prophage, found in four out of five strains isolated from kombucha tea. Int_F_ shared a higher protein sequence identity (92%) with Int_B_ and Int_C_ integrases than with the three other groups (60–61%) (Table 2). Surprisingly, the site targeted by Int_F_ prophages (*attB*_F_) was the same 63 bp *att* sequence containing the 3′ end of the *tRNA^Leu^^-^^0923^* gene, previously described as a putative alternative site for Int_B_ prophages (Figure 1). Hence, prophages with distinct integrases (Int_B_ or Int_F_) can integrate at the same *tRNA^Leu^^-^^0923^* site in the chromosome of *O. oeni*.

Finally, we used mitomycin C (MC) to induce the prophages present in a set of representative strains from the CRB Oeno collection. We first tested 11 mono-lysogens (seven Int_A_; two Int_B_; two Int_D_). We observed a clear lysis for nine strains, and the results obtained for strain IOEB0608 are shown in Appendix A. The obtained lysates formed plaques on strain IOEB277, suggesting that they contain functional phages (results not shown). The strains showing no lysis upon MC addition were IOEB0502 and C23, which harbored an Int_A_ and an Int_B_ phage, respectively. Lastly, we also tested the S28 double lysogen (Int_A_-Int_C_). The strain lysed in the presence of MC and plaques were also produced on the sensitive host. Phages were recovered from 20 individual plaques and characterized by PCR with primers targeting the *int*_A_ and *int_C_* genes [17]. Only Int_A_ phages were detected, suggesting that Int_C_ phages do not form active phage particles or that Int_A_ is the faster phage which dictates lysis time.

### 3.2. Phage Remnants Also Use tRNA Sites in O. oeni

Degeneration of prophages was observed in *O. oeni* and eight distinct phage-related genomic islands were identified. Their grounding was further supported by the observations of reduced sizes (5.2 kb to 19.5 kb), defective integrase proteins (PR_C_, PR_F1–2_, PR_I2_), the lack of other key phage genes (notably those involved in morphogenesis), the presence of transposase sequences (PR_H2_, PR_I1_) and degenerate *att* sites and/or larger sequence variability reflected by the presence of neighboring pseudogenes (PR_G_, PR_F_) (Figure 1 and Figure 2). Such phage-related genomic islands were considered as prophage remnants (PRs).

PR_G_ was rare and found in 2.6% of the tested strains. The prevalence of PR_H_ and PR_I_ elements was similar (17.7% and 16.4%, respectively). Only PR_C_ was frequently observed in the species (79.2%) (Table 1).

PRs were consistently integrated in five precise locations within the host genome, which also corresponded to tRNA genes (Figure 1 and Figure 2). Three PRs (PR_C_ and PR_F1_/_F2_) were found at the previously characterized host sites used for the SSR of Int_C_ and Int_F_ phages, respectively [17]. However, the resident PRs and prophage sequences shared no homologies. The sequences flanking the five others (PR_G,_ PR_H1/2_ and PR_I1/I2_) differed from the sites of the known prophages of *O. oeni,* and were further named *attB_G_* (OEOE_t0104; *tRNA^Lys^*), *attB_H_* (OEOE_t0530; *tRNA^Ser^*) and *attB_I_* (OEOE_t1213; *tRNA^Ser^*) (Figure 1). The length of the attachment sequences required for SSR were 61, 22 and 19 bp, respectively.

Annotation results indicated that a total of 16% of 122 ORFs found on all eight PRs were assigned to phage protein functions (Figure 2). Most were found in the larger PR_H_ regions (14.5 kb to 19.5 kb). Interestingly, only PR_F1_ and PR_F2_ still encode endolysin and holin genes. Several PR-related genes specified surface proteins of unknown function (PR_C_, PR_H1_), as well as putative phage resistance mechanisms (Sie, restriction–modification systems) (PR_F1_, PR_F2_, PR_G_, PR_I_). This suggests that PRs are not inactive DNA remnants and may encode traits useful for the host, as suggested for prophages [17]. In addition, the presence of phage resistance mechanisms in four PRs is an indicator of phage traffic and induced pressure during the making of fermented beverages.

### 3.3. Chromosomal Location and Genomic Context of Prophages and PR

As observed in many bacterial models, prophage-related regions (complete and PRs) are not randomly distributed in the *O. oeni* genome. Their presence was rare on the origin-proximal half of the *O. oeni* chromosome. Only PR_G_ was integrated near the origin of replication, where genes could be, on average, more expressed [41]. There was a significant enrichment in the absolute number of phages and PRs on the right replichore, where phage abundance increased along the *ori–ter* axis (Figure 1). Interestingly, four attachment sites (*attB*_C_, *attB*_E,_
*attB*_B_ and *attB*_F_ sites) were found to lie on a 200 kb segment of the chromosome, near *terC*. The latter region is of ultimate importance for cellular processes that interact with the chromosome shape and its organization [42,43].

As also previously reported for other bacteria [41], prophage polarization was observed in *O. oeni*. The genes of oenophages show a preference for co-orientation with the bacterial replication fork. Accordingly, Int_D_ prophages, which are on the left replichore, changed their orientation.

Phages can alter host phenotype by disrupting host genes as a result of integration. Others are known to encode cis-acting element(s) near their *attP* site that, upon integration, affect the transcription of host neighboring genes [44,45]. We therefore scrutinized the annotated genomes for all nine integration loci in *O. oeni* and retrieved the genes surrounding the *attB* sites for functional classification (average distance was 4200 bp) (Figure 1). Integration occurred next to some genes which play important roles in bacterial cellular systems, and specify proteins involved in cell wall metabolism such as the WalK/WalR two-component system [46], response to osmotic stress (mechanosensitive channel of large conductance, MscL), a quality control system monitoring protein synthesis (SsrA) or carbohydrate metabolism (glucokinase, acetoin reductase, catabolite control A (CcpA)). CcpA is involved in the adaptation to changing carbon sources. More recently, its role as a pleiotropic regulator, controlling not only carbohydrate metabolism but also stress response, has been proposed in different LAB under aerobic conditions [47]. The hypothesis that some prophages/PRs may act as switches that regulate the expression of neighboring genes is attractive in *O. oeni* and deserves further work as this may have practical significance for the production of MLF starter cultures.

Of note, two sites (*attB_C_* and *attB_F_*) could accommodate a PR, or a PR and a phage in a tandem association. We re-examined the sequences flanking these PRs and/or prophages amongst relevant strains and found that the two expected *attL* and *attR* junctions were conserved in tandem associations at the *attB_D_* site but not at the *attB_C_* locus. The latter consisted of a 78 bp repeated sequence, which may represent the ancestral *attB_C_* site. This size is larger than proposed earlier [15,17,38]. In all combinations (PR_C_ alone or tandem association with an Int_C_ prophage), we consistently identified a degenerated *attR_C_* sequence in the remnant, which may be indicative of its domestication. PR_C_ had maintained >99% nucleotide similarity across its entire genome, suggesting that it is under strong evolutionary pressure and likely provides an important biological function to the host. Its prevalence in the whole population (75.3%) was higher compared to Int_C_ phages (Table 1). It is likely that the acquisition of PR_C_ preceded that of Int_C_ phages during the evolution of *O. oeni*. The loss of the *attR_C_* site resulting from domestication did not hamper further integration of Int_C_ phages at the intact *attL_C_* site (Figure 2). In contrast, the ability of the intact Int_C_ prophages to excise is likely to be impaired as we did not observe any spontaneous or MC-induced excision of the phage in strains containing an Int_C_ phage/PR_C_ tandem association, such as S28.

### 3.4. Lysogeny Is Widespread among Wine Strains from Phylogroup A

We next explored whether quantitative and qualitative differences in patterns of prophages (as assessed from their integrase and attachment site used for SSR) differ across lineages and niches occupied by strains of *O. oeni* (Table 1). Most of the currently sequenced *O. oeni* strains in our set have, to date, been collected from wines and belong to phylogroup A (75%; *n* = 174). This enables a robust assessment of lysogeny in this particular phylogroup/niche.

Phylogroup A was particularly open to the uptake of foreign DNA of phage origin since 63% of the strains were lysogens (Table 1). Yet, prophages were not distributed uniformly across the different sub-groups described in the lineage. Interestingly, prevalence differed between the AR and AW sub-lineages, corresponding to strains adapted to red and white wines, respectively [10]. Hence, 100% of strains (*n* = 10) in sub-group AW associated with white wines produced in Burgundy and Champagne were lysogens [9]. In contrast, the proportion was only 40% (*n* = 10) in the Burgundy red wines subgroup (AR). Lysogeny was less frequent in other sub-groups of strains in phylogroup A. As an example, only one of the seven strains constituting the PSU-1 subgroup was observed to carry a prophage [9], which is lower than the mean value observed in the whole *O. oeni* species.

Phylogroup A strains appear enriched in some prophages, corresponding to members of the Int_A_, Int_B_ and Int_C_ groups. Conversely, and with few exceptions, these prophages were less abundant or absent in other lineages (phylogroups B, C and D) (Table 1). Of note, despite different origins of wine and time of collection, all 10 strains from sub-group AW (white wines) exclusively harbored an Int_A_ phage. Eight of ten had the same organization and sequence. This may reflect selection for the acquisition of locally adaptive functions that are transferred by the Int_A_ phage genomes. Alternatively, the Int_A_ phages may have piggybacked on hosts that outcompeted other variants in the process of natural selection in the specific white wines considered [48].

The second pattern seen was the frequent carriage of two distinct prophages in phylogroup A strains, as 45 out of the 47 poly-lysogens detected in the whole *O. oeni* species belonged to this particular phylogroup. This may suggest that interactions between prophages may be beneficial for the host, by reducing the rate of spontaneous lysis or regulating gene expression under specific wine conditions [27]. This hypothesis is attractive and needs more experimental support. Phylogroup A strains contained the six specific combinations of prophage carriage observed in the whole species: Int_A_ + Int_C_ (*n* = 17), Int_B_ + Int_C_ (*n* = 14), Int_A_ + Int_D_ (*n* = 8), Int_A_ + Int_B_ (*n* = 3), Int_A_ + Int_C_ + Int_D_ (*n* = 3) and Int_A_ + Int_B_ + Int_C_ (*n* = 1) (Table 1). Of note, prophages from the Int_D_ groups were found in 14 lysogens from phylogroup A, of which 11 corresponded to poly-lysogens. Since Int_D_ phages are the only prophages integrated on the left replichore, their presence and possibly low induction rate could balance the prophage integrations on the other replichore and stabilize the genome architecture. Although mono-lysogens for Int_D_ and Int_B_ prophages are found in wine strains, no Int_D_ + Int_B_ double lysogens were detected in phylogroup A, nor in the whole population. Each of these prophages may encode a resistance mechanism to phage superinfection. Alternately, the prophages may be incompatible with each other, or their presence may decrease cell fitness and lead to the extinction of the poly-lysogens in wine.

Further insight into the peculiarity of phylogroup A strains was also provided when we re-examined the distribution of PR elements (Table 1). We found that PR_C_ was present in all strains from phylogroup A, suggesting the domestication of the corresponding ancestral phage in wine strains, possibly leading to niche-specific fitness effects. These genetic signatures might also have applications as they could be used for typing purposes.

Phylogroups B, C and D are less numerously represented in our set of 231 strains (Table 1). Yet, with this caveat in mind, differences in prophage content, with respect to phylogroup A, were found. We detected prophages with lower frequency in phylogroup C (38%) while the value in phylogroup B (55%) was closer to that reported in phylogroup A (60%). Both phylogroups are known to contain strains originating from cider or wine, and the latter have been consistently isolated following completion of AF [9]. Patterns of prophages were also different in phylogroups B and C and Int_D_ phages were the most represented prophages. In particular, they corresponded to 12 out of the 15 prophages identified in mono-lysogens in phylogroup B. Only two poly-lysogens were found. Interestingly, phylogroup C was characterized by the presence of the specific Int_E_ prophages, observed in two cider strains and in strain AWRIB663 from wine. The same characteristic was observed for Int_F_ prophages, as well as PR_G_, which have, to date, been exclusively found in kombucha strains. Due to the limited number of strains in the B, C and D phylogroups, further investigation and isolation campaigns on ciders, kombucha and possibly other fruits and kefirs are now needed.

### 3.5. Integrase Phylogeny

We performed a multiple alignment of phage-related integrases from *O. oeni*. We first included a representative sequence from each of the Int_A-F_ phage groups and all four PR-encoded integrases. The size of the 10 integrases ranged from 340 to 389 aa and we found 25 invariant amino acid residues, of which 10 are located in the C-terminal catalytic domain of the phage recombinases. Amongst them, the catalytic residue tetrad Arg–His–Arg–Tyr (R–H–R–Y), which is needed for DNA cleavage and joining in the integrase family of tyrosine recombinases, was identified (Appendix A).

In order to construct a maximum likelihood tree, we next included integrases from phages infecting the related species *O. sicerae* from ciders and kefir, *O. kitaharae* from sake [29,30,31] and other LAB belonging to *Pediococcus,*
*Lactiplantibacillus*, *Fructilactobacillus* and *Streptococcus* genera [49] (Figure 3). The putative *O. oeni* XerC/XerD recombinases served as an outgroup. Of note, both XerC and XerD from *O. oeni* resemble the XerS protein which is involved in the cell recombination machinery of *Lc. lactis* [50].

As seen in Figure 3, most integrases identified in *O. oeni* belonged to two major clusters, suggesting distinct evolutionary trajectories for each group. The majority of phage integrases formed a cluster (cluster I) with sequences from phages infecting other LAB (*Lb. plantarum*, *P. acidilactici*, *Lb. casei* and *L. mesenteroides)*. This is consistent with the current literature showing the existence of cross-transmission networks between *O. oeni* and other LAB, as a variety of species coexist on grapes and must, including, notably, *Lactiplantibacillus plantarum* [13,49,51]. In addition, recent phylogenomics also resulted in the assignment of *Leuconostocaceae* and *Lactobacillaceae* into a single family [49]. A sub-group in cluster I was specific to prophages infecting species of the *Oenococcus* and *Fructilactobacillus* genera. In the latter genus, *F. fructosus* is the only species found in wine [52]. Interestingly, the Int_A_ type integrase was separate, and grouped evolutionarily with a sequence from *O. kitaharae*. Remarkably, the Int_E_ sequence grouped with the integrase found in a prophage from *O. sicereae.* The corresponding lysogens (affiliated to the *O. oeni* and *O. sicerae* species) were both isolated from cider. It would therefore be interesting to assess whether the oenophages can infect individual hosts from different species in cider and other fermented beverages. We also observed that the three integrases Int_B_, Int_C_ and Int_F_, which drive the integration of their cognate phages in the same region in the host chromosome, were more closely related (Figure 1). Lastly, the only integrase from a remnant in cluster I was that from PR_F_. In contrast, the integrases harbored by PR_H1-H2_, PR_I1-I2_ and PR_G_ formed a distinct lineage, and clustered with two integrases associated with other LAB species. The inspection of the alignment showed that these five PR-associated integrases had, notably, an additional domain of 20 amino acids upstream of the tyrosine catalytic residue compared to other oenophage integrases. They may not represent remnants of previous lysogenization by full-length prophages, but rather belong to a unique family of mobile genetic elements.

### 3.6. Correlation of Phage Phylogeny with attB Location

The correlation of phage integrase phylogeny with *attB* location on the bacterial chromosome was confirmed with two notable exceptions, which were both related to the occupancy of the *attB*_F_ site (Figure 1 and Figure 3). This particular locus, close to the replication termini, could indeed accommodate an Int_B_ prophage, or an Int_F_ prophage and/or a PR_F_ remnant in the different strains of our panel. This raised two interesting questions. How did the Int_B_ recombinase display a relaxed specificity, driving the integration of the phage genome (hereby an Int_B_ group member) at distinct locations (*attB*_B_ or *attB*_F_) in the chromosome of distinct strains of *O. oeni*? Next, we also questioned the reason why PR_F_ integrates at the *attB*_F_ site, although its cognate integrase was more closely related to members of the Int_D_ than to the Int_F_ group. The rationale behind these observations and questions was that a more detailed analysis of the SSR units of these oenophages and PRs was needed.

#### 3.6.1. SSR of Int_F_ and Int_B_ Prophages

During SSR, cross-activity of a recombinase of one phage with the attachment site of another is proportional to the degree of homology between their integrases and similarity between the core- and arm-type sequences in attachment sites of the respective phages [53]. We therefore clarified the site preference pattern and sequence requirements for the SSR of Int_F_ and Int_B_ phages by analyzing their *attP* sites. To carry out the analyses, phage attachment core sequences were deduced from the *attL* and *attR* junctions within the host chromosome identified in lysogens (accessed from GenBank) and from the *attP* sequences from the genome of free Int_B_ phages isolated from the enological environment [18,54]. For clarity, the phages belonging to the Int_B_ group were subdivided into the Int_B-F_ and Int_B-B_ sub-groups, depending on the attachment site targeted in the bacterial chromosome (*attB_B_* or *attB_F_*). Prophages harbored by strain LAB2013 and IOEBB10 were their representatives, respectively. Int_F_ phages were represented by the prophage of strain UBOCC315001.

SSR of Int_F_ phages was found to complement the 13 bp 3′ end of the bacterial *tRNA^Leu 923^*. This sequence does not encompass the tRNA anticodon loop. The identity block common to the phage and bacterial sequences extended well beyond, and a 63 bp homologous pair was identified (Figure 4A). It is well established that SSR requires a longer sequence at *attP* than at *attB*. Tyrosine recombinases usually possess a low binding affinity to the core site of *attP* and a high binding affinity for the flanking arm regions [53]. We next questioned what sequences within *attP_F_* actually support integrase binding in *O. oeni*. We addressed this question by screening the arm regions for direct repeats (DRs), which could serve as putative arm-binding sites recognized by the N-terminus of the integrase (Figure 4A). A short 10 bp DR was found (5′ATTTGCACAA3′, Figure 4A). It was not evenly distributed on both arms around the core, with two and four repeats on the left and right arms (P1 to P6), respectively (Figure 4A and Table 3). Like other characterized systems amongst LAB phages, the proposed core site exhibited essentially no symmetry and no putative inverted core-binding sites were identified [55]. With the caveat that additional putative binding sites for accessory proteins (excisionase, integration host factor) may be present, we propose that the size of the *attP* sequence of Int_F_ phages is at least 240 bp. This size is consistent with the well-documented model proposed by Campbell for λ integration in *Escherichia coli* [56,57].

Recombination of Int_B-F_ prophages also required the same 63 bp core sequence, common to *attP* and *attB_F_*. However, the detailed analysis of the whole *attP*_B-F_ region showed some deletions, resulting in the loss of the P4 repeat on the right arm (Figure 4A, Table 3). As a consequence, there was a reduction in the number of DRs, as well as a modified spacing between the P3 and P5 repeats, which may impact protein–protein and/or protein–DNA interactions during intasome formation. Another striking difference was found when the Int_B-F_ and Int_F_ protein sequences were compared (Appendix A). Despite their high identity (92%), they differed by 46 amino acid (aa) changes, of which 21 (45%) mapped to the 66 aa amino terminus, representing 18% of the integrase. Similar results were observed with the Int_B_ integrases obtained from the free replicating phages OE33PA and B148 previously isolated from wine [17,18]. The size of the Int_F_ N-terminus domain which concentrates the mutations (66 aa) is in agreement with the assignment of the minimal Int fragment of 64 aa that binds to arm-type sites in the lambda model [58].

Altogether, our data are consistent with the existence of different N-termini in Int_F_ and Int_B-F_ integrases, allowing each recombinase to bind the specific sets of DRs on the arm regions in each *attP* site. The observed mutations may have a broader impact on SSR since the N-terminal domain of Int also plays a role in modulating the activity of core-binding and catalytic domains [59]. In addition, the impact of the mutations associated with the other domains of the integrase protein sequences also needs further work.

Noteworthily, Int_F_ and Int_B-F_ phages were associated with strains from distinct niches, corresponding to kombucha and wine, respectively. It is likely that the presence of slight modifications in their SSR units needs to be examined in light of the distinct niches where their bacterial hosts are evolving. Our observations may reflect the coevolution of the *attP_F_* and integrase sequences in response to changing conditions. They may be a signature of distinct evolutionary trajectories across the different niches colonized by *O. oeni*, and represent signatures of adaptation to wine. We posit that modifications in the *attP_F_* site may have progressively arisen during the adaptation of *O. oeni* to more drastic variations in habitat conditions (wine) and have been coupled to mutations in the Int_F_ sequence. The latter adapted the novel integrase (Int_B_) to this new and altered site, resulting in the variations observed within the SSR units of Int_F_ and Int_B-F_ phages [60]. Altogether, our observations may reflect the evolutionary processes resulting in niche divergence. If this hypothesis is correct, it questions the existence of selection for the preservation of *attB_F_* as an integration site for phages in two distinct niches (kombucha and wine).

Last, it is likely that Int_B-F_ phages went through additional and distinct evolutionary processes in wine, yielding Int_B-B_ phages. Hence, the latter prophages were shown to integrate into a distinct gene, 66 kb away from the *attB_F_* site used by Int_B-F_. Both sites were part of *tRNA^Leu^* genes. The SSR of Int_B-F_ and Int_B-B_ phages required the same overall region in the phage genomes, but the nature of the sequences involved were different. Striking differences found in Int_B-B_ phages included a reduced number of DRs on arms (5) and the substitution of the 5′ extremity of the 63 bp sequence by a new 9 bp sequence. In addition, the identity block between host and Int_B-B_ phage sequences was reduced to the first 15 bp of the previously proposed 63 bp core (Figure 4B). Since the Int_B_ integrase recombines regardless of the lateral 9 bp sequence in the *attP* site present in Int_B-B_ and Int_B-F_ phages, the latter nucleotides probably do not represent core-binding sequences for the integrase. The role is probably devoted to the downstream 6 bp sequence that is common to *attP_B-B_* and *attP_B-F_* sites and where cleavage is likely to occur. The emergence of Int_B-B_ phages is likely to result from mutations leading to a much more substantial increase in the recognition of a secondary site by Int_B-F_ phages followed by an abnormal excision. This hypothesis is consistent with the chromosome jumping model described in lambda [61].

#### 3.6.2. SRR of PR_F_ Elements

The PR_F1_ and PR_F2_ remnants are integrated in the *attB_F_* site in strains collected from cider and wine and from kombucha (phylogroups B and D, respectively). Yet, their integrase was more related to the Int_D_ than to the Int_F_ type (Figure 3). Both integrase-coding gene were split into two ORFs in PR_F1_ and PR_F2_, possibly due to a frameshift mutation (Figure 2). The sequence was manually inspected and a single nucleotide was added in the premature stop codon to suppress the nonsense mutation and the corrected deduced protein sequence was used to build the phylogeny of integrase proteins (Figure 3). We assessed whether an error at this step may have caused an alignment bias, explaining the incongruence of the PR_F_ recombinase in the tree. To verify this, we compared the three *int* nucleotide sequences found in PR_F_, Int_F_ and Int_D_ prophages (Appendix A). This confirmed the closer relation of the integrase nucleotide sequences from PR_F_ and Int_D_ prophages. The reason for the discrepancy between the integrase type and integration site of PR_F_ is not known. However, it can be suggested that PR_F_ elements have resulted from modular exchanges in the lysogeny modules between two integrated Int_D_ and Int_F_ oenophages, due to intra-chromosomal homologous recombination. Homologous recombination between two prophages integrated equidistant from the *ter* region has been reported in *S. pyogenes*. Such phage-related rearrangements resulted in a large chromosomal inversion of the region between the attachment sites, and the emergence of two novel hybrid prophages with exchanged genes [62]. In *O. oeni*, *int_D_* and *int_F_* sequences contain homologous regions. In addition, such homologous recombination events may also be mediated by the bacterial sequences flanking each prophage which both correspond to a *tRNA^Leu^* gene.

### 3.7. Most attP Regions in Oenophages Derive from Two Distinct Sequences

Surprisingly, all *attP* sequences upstream of the integrase gene in Int_A_, Int_B_, Int_C_, Int_D_ and Int_F_ prophages had similar sequences, suggesting a common origin (Figure 5).

Despite the presence of several indel events, all core sequences involved in RSS (underlined in Figure 5) had a common feature and were flanked by direct repeat sequences, which are proposed to represent the binding sites for the different recombinases (Table 3). Coevolution of the phage components of SSR units have progressively occurred and involved indels in the core, the slippering of DR sequences and modifications in the N-terminus of the integrase (Figure 5). Through such events, recombinase activity is retained while the core *attP* sites are progressively adjusted to novel loci in the chromosome of the host. This may indicate long-term coexistence between these phages and the host in fermented beverages.

Of note, the *attP*s from Int_E_ phages were distinct in their sequence and also lacked DR repeats compared to other oenophages. Future studies should experimentally confirm whether the Int_E_ recombinase can utilize the candidate core sequences for recombination reactions without flanking inverted repeats. The comparative analysis of a set of completely sequenced oenophage genomes has recently demonstrated that prophages are distributed into two clusters of *cos* (members of Int_A_ and Int_B_ groups) and *pac* (Int_D_) phages [63]. Even though the SSR unit of Int_E_ phages has some peculiarities, the latter have not evolved completely independently. As seen in Figure 6, Int_E_ phages are closely related to the *cos* phages, together with Int_C_ and Int_F_ prophages. Corresponding genomes in this cluster are mosaics, whereby individual phages are constructed as assemblages of modules, many of which are single genes (result not shown). Of note, this cluster harbors prophages which are integrated on the same replichore in the chromosome of *O. oeni*. In contrast, the cluster corresponding to Int_D_ prophages is less diverse and members share relatively few genes with the other cluster. 

## 4. Concluding Remarks

A short minimal doubling time under optimal growth conditions was shown to represent the trait most correlated with lysogeny among bacteria, and the frequency of lysogeny was also increased with bacterial genome size [41]. Clearly, *O. oeni* stands out from the well-studied model systems used in this study, and probably faces specific ecological conditions which constrain the lytic–lysogeny decision and favor lysogeny and poly-lysogeny. This may generate more benefits than costs for both partners.

Our results also confirmed that tRNA genes are the preferred chromosomal integration sites in *O. oeni* and that integration tropisms are associated with the phylogeny of the phage integrases. Interestingly, all bacterial attachment sites, except *attB_D_*, are located in one replichore (one half of the chromosome) and four lie within a 300 kb region, which is therefore a region of high plasticity in the species. Such lopsided phage integrations into chromosomal DNA may result in an unsymmetrical genome architecture across the replication axis, and induce chromosomal rearrangement for stabilizing the genome architecture. Yet, such rearrangements are not observed in *O. oeni* [9].

The different niches where the *Oenococcus* species is found have unique physical, chemical and biological profiles that likely promote speciation of both phages and their bacterial hosts. Accordingly, phages infecting strains associated with kombucha, cider and wines were observed to exhibit differences in their SSR units and more work is now needed to explore their gene repertoire as well, and assess whether they provide the bacterial hosts with additional genes and competitive advantages.

Considering whether prophages (or combinations thereof) have a positive role in the adaptation of *O. oeni* to wine, it could be expected that strains would domesticate such beneficial prophages, as seen with PR regions, in order to prevent their excision. This is probably the case for Int_C_ prophages, as we could not yet detect excision of phage DNA/particles in past studies. In contrast, many pieces of experimental evidence show that members of the frequently encountered Int_A_ and Int_B_ prophages in wine are still active and can excise. Therefore, it is unclear why Int_A_ and Int_B_ phages are less prone to grounding, compared to members of the Int_C_ group. Reversible lysogeny could be of importance for genome architecture, as the grounding of the various prophages located on the right replichore could be detrimental. In addition, the ability of such prophages to excise could represent a competitive mechanism to eliminate sensitive non lysogenic strains, or to lysogenize them, increasing the tolerance of the population to stressful conditions. Conversely, site-specific integration of Int_A_ and Int_B_ phages may be beneficial under certain circumstances and modulate the expression of essential genes in stressful conditions. These characteristics might reflect the different evolutionary strategies and opposite selection pressures as a consequence of adaptation to diverse niches in which the different phylogroups have evolved.

## Figures and Tables

**Figure 1 microorganisms-09-00856-f001:**
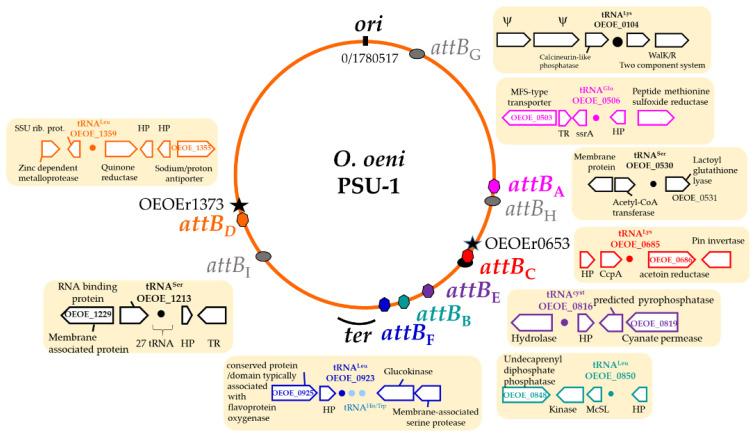
Site-specific recombination of temperate oenophages in *O. oeni.* The locations of the attachments sites used by prophages (color) and phage remnants (gray) are shown as a generalized chromosome backbone based upon the PSU-1 genome. Locations of the *oriC*, *Ter* site-containing region and rRNA loci (stars) are indicated. Genes flanking all *attB* sites are indicated.

**Figure 2 microorganisms-09-00856-f002:**
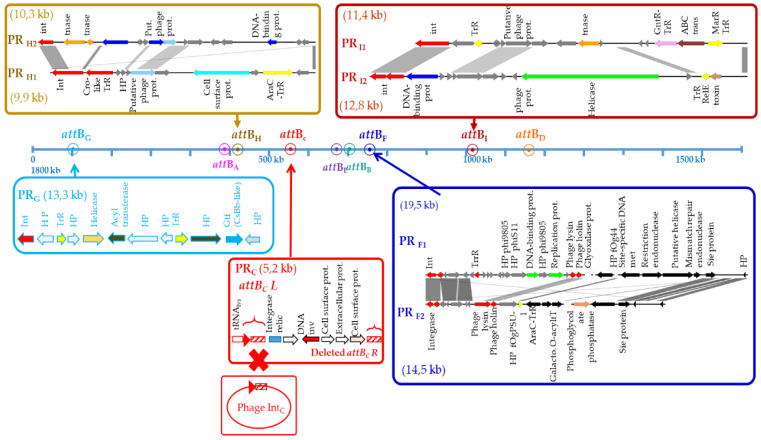
Localization and genetic organization of prophage remnants in the genome of *Oenococcus oeni*. Genes encoding hypothetical proteins are represented in gray.

**Figure 3 microorganisms-09-00856-f003:**
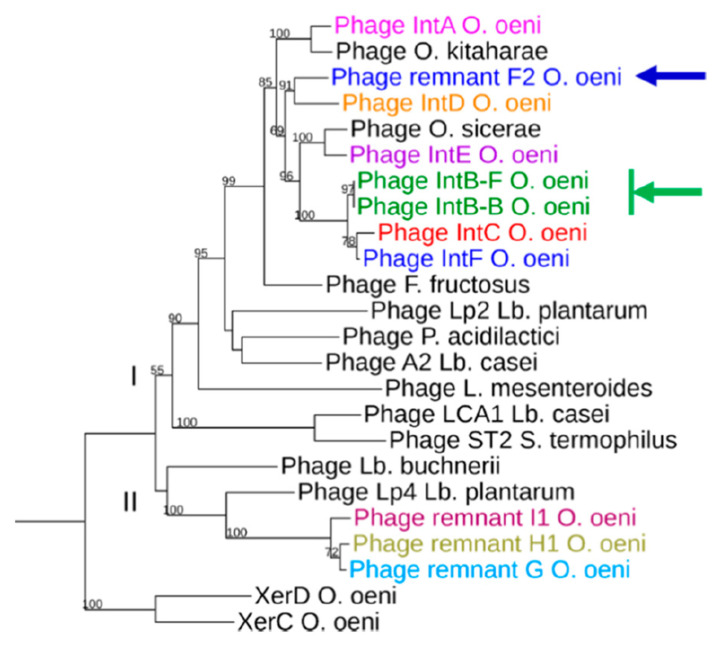
Phylogenetic tree of integrases found in *O. oeni* and other LAB. Sequences are distributed in clusters I and II. Their origin is as follows: Int_A_ (IOEB0608); Int_B_ Φ10MC (AAD00268); Int_C_ (IOEBS28); Int_D_ (IOEB9805); Int_E_ CRBO1384; Int_F_ (UBOCC315001); Int PRI_1_ (IOEBS277); Int PRH_1_ (AWRIB422); Int PR_G_ (UBOCC315001); Int PRF_1_ (DIV5-23); Int from phages Lca1 (ABD83428) and A2 (NP_680502); unnamed phage infecting *P. acidilactici* 7-4 (ZP_06197614); Int from prophages found in *Lb. plantarum* WCFS1 (NP_785910), STIII (ADN97941) and Lp2 prophage (WP_011101785.1); *O. sicerae* (WP_128686695.1); *O. kitaharae* (EHN59022); *L. mesenteroides* ATCC19254 (ZP_03913849) and *Lb. buchneri* ATCC 11577 (ZP_03943244). The XerC and XerD proteins were obtained from *O. oeni* PSU-1 (WP_002817147 and WP_002820463.1) and served as an outgroup. Arrows represent incongruences between integrase phylogeny and *attB* location of the corresponding prophages in the bacterial genomes. The PRF_2_-associated integrase-coding gene was split into two ORFs and carried a frameshift mutation due to a missing nucleotide. A single nucleotide was added in the premature stop codon to suppress the nonsense mutation.

**Figure 4 microorganisms-09-00856-f004:**
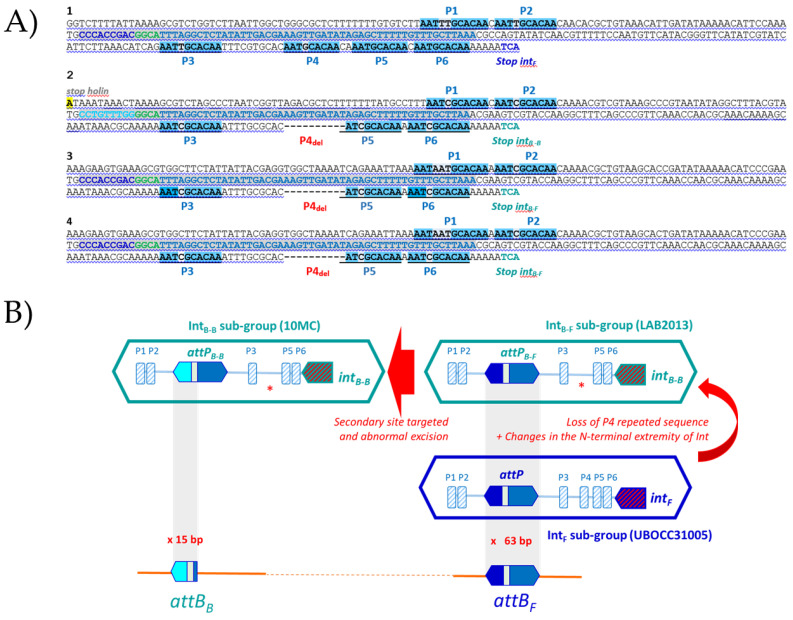
Interrogation of oenococcal genomes for the SSR units of Int_B-B_, Int_B-F_ and Int_F_ prophages. (**A**) Alignments of the *attP* regions involved in SSR for Int_F_ phages (1 in blue), the proposed sub-types Int_B-B_ and Int_B-F_ phages (3 and 4 in green). 1, Int_F_ prophage from strain UBOCC315005; 2, Int_B-B_ prophage from strain IOEBB10; 3, Int_B-F_ prophage from strain LAB2013; 4, Int_B-F_ phage OE33PA. P1 to P6, Direct repeats. The 63 bp block identity is in gray. (**B**) Boxed sequences represent the *attP* sequences of Int_B_ and Int_F_ phages. Arrows indicate the possible complex mechanisms of evolution of SSR units during niche adaptation.

**Figure 5 microorganisms-09-00856-f005:**
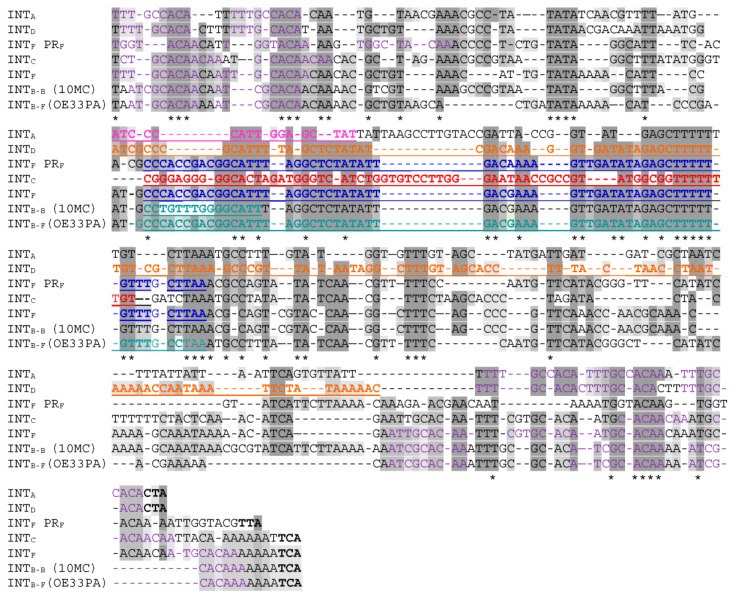
Alignments of the *attP* sequences found in the different oenophages. The homologous core sequences involved in recombination with the bacterial chromosome are underlined and in color (the color code is the same as shown in Figure 1). All direct repeats serving as putative binding sites for integrases are in purple. The three last nucleotides correspond to the stop codon of the integrase genes. A clustal alignment was first constructed and gaps were introduced manually.

**Figure 6 microorganisms-09-00856-f006:**
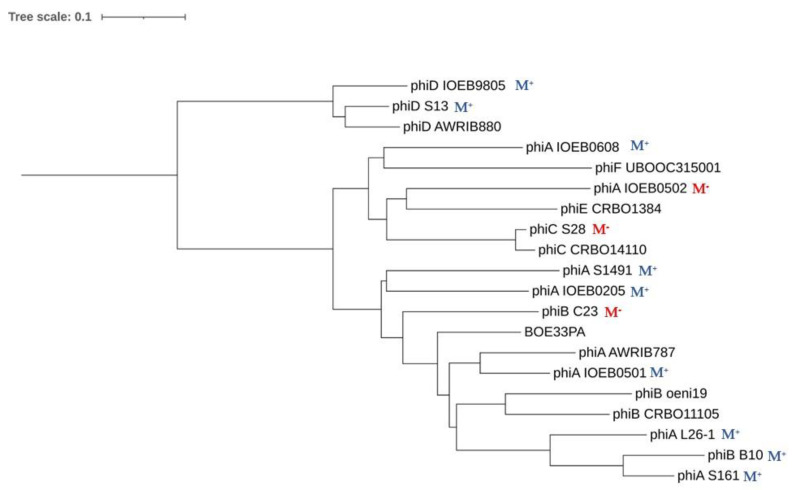
Phylogenomic Genome BLAST Distance Phylogeny (GBDP) tree of representative oenophages from the CRB Oeno Collection. The tree was generated by VICTOR and visualized with Fig Tree [36]. Int groups are represented by letters (A to F). B10 is the prophage found in strain IOEB B10. M^+^ and M^−^ indicate the presence or absence of (i) a lysis when the lysogen was grown in the presence of mitomycin C and (ii) plaques when the MC lysate was tested on the sensitive strain IOEB S77.

**Table 1 microorganisms-09-00856-t001:** Lysogeny in a set of 231 strains representing the four described phylogroups of *O. oeni* and different combinations of prophage carriage.

Distribution and Type of Prophages	Phylogroups in the *O. oeni* Species			
	A	B	C	D
Number of strains analyzed	174	31	21	5
Number of mono- and poly-lysogens	105	17	8	4
Lysogeny (%)	60	55	38	80
Mono-lysogens (*n* = 86) and prophage types				
Int_A_	22	2	0	0
Int_B_	14	1	1	0
Int_C_	20	0	0	0
Int_D_	3	12	4	0
Int_E_	0	0	3	0
Int_F_	0	0	0	4
Poly-lysogens (*n* = 48) and prophage combinations				
Int_AB_	3	1	0	0
Int_AC_	17	0	0	0
Int_AD_	8	0	0	0
Int_BC_	14	0	0	0
Int_ABC_	1	1	0	0
Int_ACD_	3	0	0	0
Phage remnants (PRs) in strains				
PR_C_	174	0	9	0
PR_F_	0	1	1	4
PR_G_	0	0	0	4
PR_H_	18	6	12	5
PR_I_	9	14	15	0

**Table 2 microorganisms-09-00856-t002:** Nucleotide and amino acid identities in pairwise comparisons amongst the individual genes of phage integrases in *O. oeni*.

Integrase Types	Int_A_	Int_B_	Int_C_	Int_D_	Int_E_
Int_B_	69 ^a^/62 ^b^				
Int_C_	68/60	84/86			
Int_D_	74/66	68/61	68/59		
Int_E_	66/61	72/69	72/68	66/60	
Int_F_	70/61	88/92	88/92	69/60	73/68

Representative prophage integrase sequences for each group were obtained from the following *O. oeni* strains: Int_A_ from IOEB0608, Int_B_ from IOEBB10, Int_C_ from IOEBS28, Int_D_ from IOEB9805, Int_E_ from CRBO1384 and Int_F_ from UBOCC315001. ^a^ means: nucleotide identity. ^b^ means amico acid identity.

**Table 3 microorganisms-09-00856-t003:** Characteristics of the *attB* and *attP* regions from oenophages and phage remnants.

Phage-Related Elements			Host Strain and *attB* Site		Putative Integrase Binding Sites (DR *) on *attP*		
P/PR	Strain Harboring the Prophage or PR Name	Int	Name, Phylogroup and Niche	*attB* Site (bp)	nb	Right/Left Arms	DR Sequences
P	IOEB0608	Int_A_	IOEB0608 (A/wine)	*attB_A_* (tRNA^Glu^, 16 bp)	5	2/3	-TT**T**-**G**C**CACA** (x2) -TT**T**-**G**C**CACA** (x2) -TT**T**-**G**C**CACA** (x1)
	10MC	Int_B-B_	IOEB B10 (A/wine)	*attB_B_* (tRNA^Leu^, 15 bp)	5	2/3	--A**T**C**G**-**CACA**A (x5) --A**T**C**G**-**CACA**A (x1)
	LAB2013	Int_B-F_	LAB2013 (A/wine)	*attB_F_* (tRNA^Leu^, 63 bp)	5	2/3	--A**T**-**G**-**CACA**A (x1) --A**T**C**G**-**CACA**A (x3) --A**T**C**G**-**CACA**A (x1)
	IOEBS28	Int_C_	IOEB S28	*attB_C_* (tRNA^Lys^, 78 bp)	4	2/2	--A**T**-**G**-**CACA**ACAA (x3) --C**T**-**G**-**CACA**ACAA (x1)
	IOEB9805	Int_D_	IOEB9805 (B/wine)	*attB_D_* (tRNA^Leu^, 128 bp)	5	2/3	-TT**T**-**G**-**CACA** (x5)
	CRBO1384	Int_E_	CRBO1384	*attB_E_* (tRNA^Cyst^, 11 bp)	4	2/2	**T**-**G**C**CAC**-CGTT (x4)
	UBOCC315005	Int_F_	UBOCC315005 (D/kombucha)	*attB_F_* (tRNA^Leu^, 63 bp)	6	2/4	ATT**T**-**G**-**CACA**A (x1) AT-**T**-**G**-**CACA**A (x2) AA-**T**-**G**-**CACA**A (x3)
PR	PR_C_	Int_C_	IOEB S28	*attB_C_* (tRNA^Lys^, 78 bp)	4	2/2	--A**T**-**G**-**CACA**ACAA(x3) --C**T**-**G**-**CACA**ACAA (x1)
	PR_F1_	Int_F1_	DIV5-23 (B/cider)	*attB_F_* (tRNA^Leu^, 63 bp)	5	3/2	---**T**G**G**-T**ACA**A (x4) ---**T**G**G**CT**ACA**A (x1)
	PR_F2_	Int_F2_	CRB01381 (C/cider)	*attB_F_* (tRNA^Leu^, 63 bp)	6	3/3	---**T**G**G**-T**ACA**A (x4) ---**T**A**G**CT**ACA**A (x1) ---**T**G**G**-T**CAA**A (x1)
	PR_G_	Int_G_	UBOCC315005	*attB_G_* (tRNA^Cyst^, 61 bp)	nf		
	PR_H1_	Int_H1_	AWRIB422	*attB_H_* (tRNA^Ser^, 22 bp)	nf		
	PR_H2_	Int_H2_	IOEBC52	*attB_H_* (tRNA^Ser^, 22 bp)	nf		
	PR_I1_	Int_I1_	IOEB277	*attB_I_* (tRNA^Ser^, 19 bp)	nf		
	PR_I2_	Int_I2_	AWRIB576	*attB_I_* (tRNA^Ser^, 19 bp)	nf		

* DR, Direct repeat; nf, not found.

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
