# Peer review of "Distribution of Prophages in the Oenococcus oeni Species"

_microorganisms, 2021, doi:10.3390/microorganisms9040856_

Round 1
Reviewer 1 Report
The manuscript written by Claisse et al is a very good example of innovative work. Although the existence of bacteriophages in O. oeni is a real problem in enology, the last 25 years have been offered the scientific community the possibility to assess those perspectives. The work developed by the Univesité de Bordeux should be published in Microorganisms. To be honest, I can not add suggestions, as I think the present form is very satisfactory. Maybe some italics should be checked as O. oeni should be always O. oeni. Congratulations for your magnific contribution.
Author Response
Dear reviewer one,
Thank you for your comments. Italics have been checked thoughout the manuscript
Reviewer 2 Report
The study conducted by Claisse et al. brings new insights regarding the presence of prophages in the species of Oenococcus oeni from different niches. The work is well written and mainly based on in silico analyzes that revealed the distribution and the main insertion sites used by bacteriophage(s) within the host genome. As a main result, the authors showed that six different insertion sites are frequently used for site-specific recombination and that this distribution is directly related to the niche in which these bacteria were isolated.
The main weakness of this study regards the absence of prediction, distribution, and discussion of the CRISPR-Cas system (included spacers) found in the bacterial isolates used in this study (231 in total). In my point of view, this analysis would strengthen this study. Besides, a greater description of which genes these prophages carry on and whether anti-phage defense mechanisms are present is poorly explored (again CRISPR-Cas and RM system). The families to which these phages belong are not mentioned and discussed. Lastly, the software used to draw the figures are not appropriately mentioned.
In the attached document the authors can find my main concerns.

Author Response
see letter enclosed

Round 2
Reviewer 2 Report
Dear authors,
The manuscript has been significantly improved and all my concerns have been addressed.
Congratulations on your work!